# A Comparative Study of the Fitness and Trueness of a Three-Unit Fixed Dental Prosthesis Fabricated Using Two Digital Workflows

**Daehee Jang [1,†], Keunbada Son [2,3,†]**  **and Kyu-bok Lee [1,3,*]**

1   Department of Prosthodontics, School of Dentistry, Kyungpook National University, 2177 Dalgubeol-daero, Jung-gu, Daegu 41940, Korea
2   Department of Dental Science, Graduate School, Kyungpook National University, 2177 Dalgubeol-daero, Jung-gu, Daegu 41940, Korea
3   Advanced Dental Device Development Institute, Kyungpook National University, 2177 Dalgubeol-daero, Jung-gu, Daegu 41940, Korea
*   Correspondence: kblee@knu.ac.kr; Tel.: +82-053-600-7674
†   These authors contributed equally to this work (co-first author).

**Abstract:** The purpose of this study was to measure and correlate the fitness and trueness of a 3-unit fixed dental prosthesis (FDP) fabricated using two digital workflows. The 3-unit FDPs were fabricated using two digital workflows ($N = 15$). The digital workflows were divided into chairside (closed type) and in-lab (open type) groups. The scanning, computer-aided design (CAD), and computer-aided manufacturing (CAM) processes were conducted with 3shape E1 scanner, exocad CAD software, and DDS EZIS HM, respectively, in the in-lab group; and with CEREC omnicam intraoral scanner, CEREC CAD software, and CEREC MC XL, respectively, in the chairside group. The fitness of the fabricated 3-unit FDPs was evaluated by scanning the silicone replica of the cement space and analyzing the thickness of the silicone replica in the three-dimensional (3D) inspection software (Geomagic control X). The trueness of the milling unit was analyzed by 3D analysis of the CAD reference model, which is the design file of the 3-unit FDP, and the CAD test model, which is the scanned file of the 3-unit FDP. In the statistical analysis, comparison of the two groups was conducted by Mann–Whitney U test, and the correlation between the fitness and trueness was conducted by Pearson correlation test ($\alpha = 0.05$). The marginal and internal fit were significantly lower in the in-lab group at all measurement positions ($p < 0.001$). The trueness of the milling unit was significantly higher in the in-lab group compared to the chairside group ($p < 0.001$). There was a positive correlation between the trueness and internal fit (correlation coefficient = 0.621) in the in-lab group ($p = 0.013$). The use of appropriate equipment in an in-lab (open type) digital workflow enables a better fabrication of 3-unit FDPs than a chairside (closed type) digital workflow, and poor trueness on the inner surface of the crown adversely affects the internal fit.

**Keywords:** marginal and internal fit; trueness; dental CAD-CAM; dental ceramic; dentistry

## 1. Introduction

With the introduction of the computer-aided design and computer-aided manufacturing (CAD–CAM) system, fixed dental prostheses (FDP) can be produced using a digital workflow through three-dimensional (3D) data acquisition, CAD, and CAM processing, as opposed to the conventional workflow involving a manual process using the lost wax technique [1–3]. Prior to the introduction of intraoral scanners, the digital workflow was used to fabricate a gypsum model through impression-taking in the patient's mouth, and to obtain a virtual model using a desktop optical scanner [2,4]. On the other

hand, the intraoral scanner can acquire a virtual model through direct optical scanning of the patient's mouth. The introduction and development of intraoral scanners have led to a significant increase in the demand for digital workflows [5–7].

The digital workflow for FDP can be divided into a partially and fully digital workflow [8]. The partially digital workflow is used in combination with a conventional workflow for impression-taking and gypsum model-making [8]. On the other hand, a fully digital workflow does not require the production of a gypsum model because it directly scans the patient's mouth in the chairside environment [8]. The application of a fully digital workflow is increasing [9]. However, due to the cost burden of intraoral scanners and the need to fabricate the FDP in dental laboratories, a partially digital workflow may be preferred [10].

The compatible equipment used in digital workflows differs according to each production process (scan acquisition process, CAD process, and CAM process) and the manufacturer [11]. As the demand for a digital workflow increases, some manufacturers support accessibility to other manufacturers, while other manufacturers highly support self-compatibility. Equipment allowing accessibility with other manufacturers have the advantage of being usable in combination with a wider variety of equipment desired by users in each manufacturing step. On the other hand, equipment with closed accessibility has the advantage that they can be easy to control in each manufacturing step.

Many previous studies have validated FDP by evaluating their marginal and internal fit [12–18]. Poor marginal fit is a known cause of periodontal disease, cement dissolution, and limits long-term healthy use [12,16]. The clinically acceptable range of marginal fit proposed by many studies was reported to be within 120 μm [12,16–18]. An ideal internal fit increases the retention of the FDP [12,16]. Thus, many studies have examined the marginal and internal fit.

In the digital workflow, virtual FDP designed in the CAD software is produced by actual FDP through CAM processing. Many studies have been carried out to analyze the machining accuracy of the CAM process by comparing the designed virtual FDP with the actual FDP [19–21]. The ceramic FDPs are typically fabricated through the CAM process by a subtractive (milling) process using zirconia and lithium disilicate materials [22–24]. Poor trueness of the milling unit makes it difficult to reproduce the designed FDP, which can affect the marginal and internal fit. Although studies have examined the trueness and fitness, studies on the relationship between the trueness of the milling unit and the FDP fitness are lacking.

According to a systematic review by Russo [25], multi-unit FDP fabricated using the digital workflows have been reported to require further studies for clinical reliability. Therefore, the purpose of this study was to evaluate and correlate the fitness and trueness of a 3-unit FDP fabricated from two digital workflows. The null hypothesis is as follows: (1) The 3-unit FDPs fabricated from two digital workflows have no difference in their trueness and fitness; (2) the fitness and trueness evaluated in the 3-unit FDP fabricated from two digital workflows are not correlated.

## 2. Materials and Methods

### 2.1. Sample Preparation

Pilot experiments were conducted five times to determine the sample size and 12 samples were calculated using a power analysis software (G*Power v3.1.9.2, Heinrich Heine University, Düsseldorf, Germany) (actual power = 99.43%; power = 99%; $\alpha$ = 0.05). To increase power, the number of samples presented in this study was determined to be 15.

Figure 1 shows the experimental design. A maxillary resin typodont (AG-3 ZPVK, Frasaco GmbH, Tettnang, Germany) was prepared for a 3-unit FDP involving the first premolar and first molar. The margin was prepared with a depth of 1.2 mm in the chamfer form and the angle was set at 6°. The prepared typodont was scanned using a desktop optical scanner (E1 scanner, 3Shape, København, Denmark) and the scanned model was fabricated using a 3D printer (FS271M, Farsoon Technologies, Hunan, China) using Co-Cr materials. The metal model was then polished (Figure 2A).

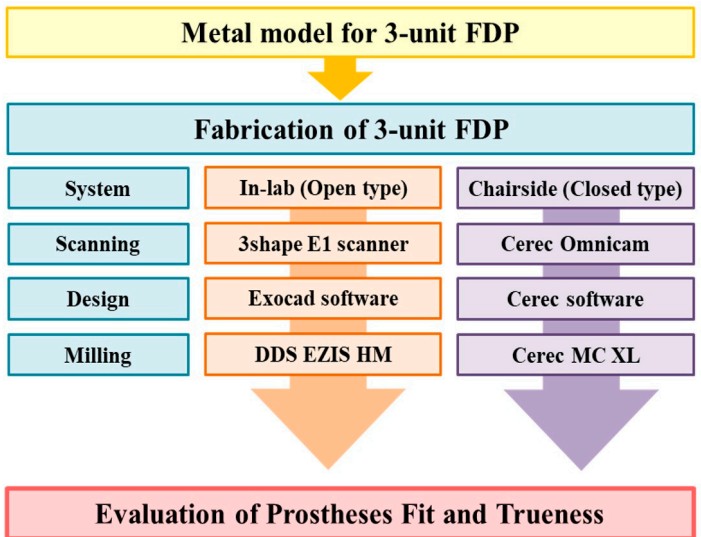

**Figure 1.** Experimental design.

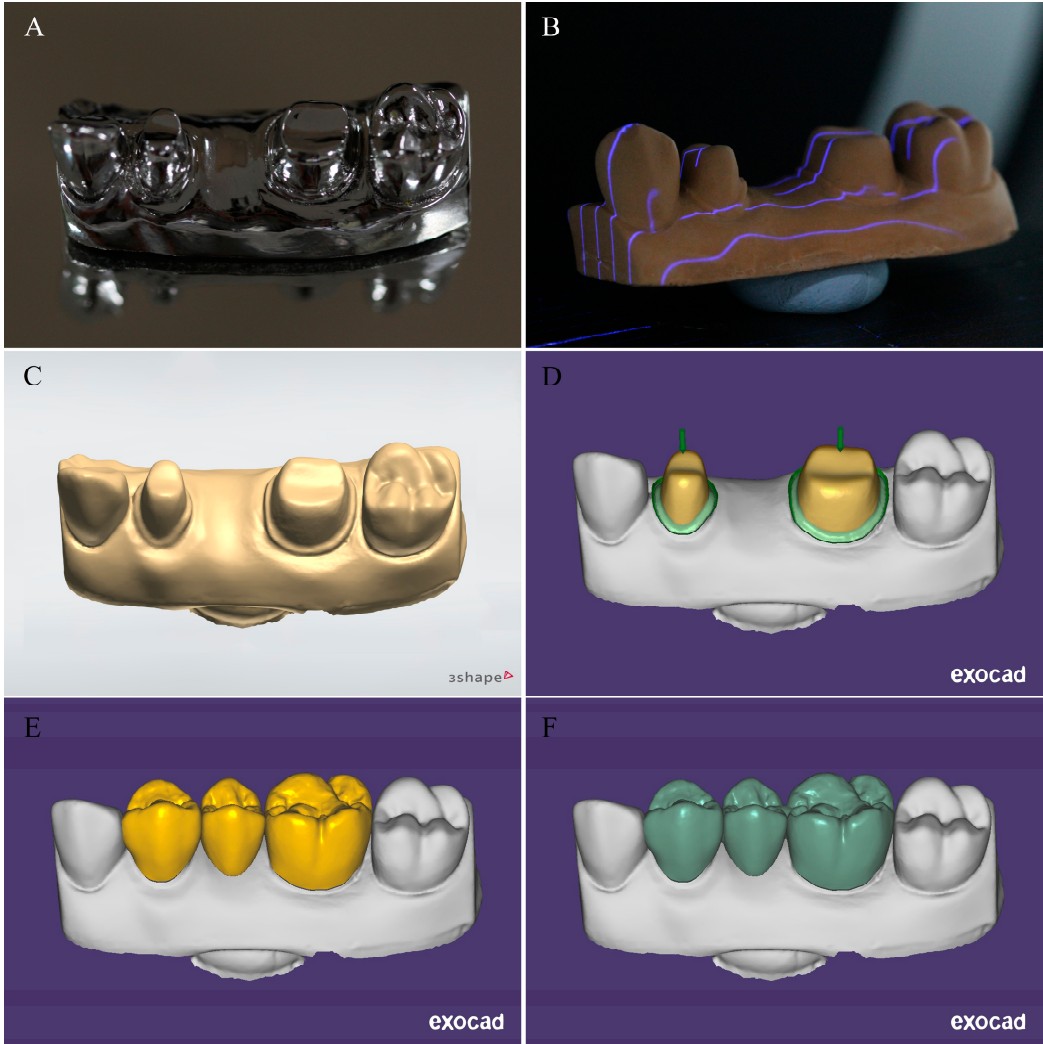

**Figure 2.** Scan process of model and design process of three-unit fixed dental prosthesis (3-unit FDP). (**A**) Metal model. (**B**) Scan process of model. (**C**) Virtual model. (**D**) Design process: Parameter setting of cement space. (**E**) Design process: Modification of prosthesis size and position. (**F**) Design process: Virtual 3-unit FDP.

The 3-unit FDPs were fabricated by the in-lab (open type) and chairside (closed type) groups according to the digital workflow. The in-lab (open type) group used a partially digital workflow. An impression of the metal model was taken by simultaneously injecting light body impression materials (Aquasil Ultra Rigid, Dentsply Sirona, York, PA, USA) around the abutment along with heavy body impression materials (Aquasil Ultra Rigid, Dentsply Sirona) on a custom impression tray. Type IV stone (Fujirock, GC, Leuven, Belgium) was poured into the impression and stone casts were made. The stone casts were used to obtain a virtual model by using the desktop optical scanner (E1 scanner, 3Shape) (Figure 2B,C). Prior to use, the desktop optical scanner was calibrated according to the manufacturer's recommendations. The acquired virtual models were transferred to the CAD software (exocad, exocad GmbH, Darmstadt, Germany), which was used to design the 3-unit FDPs (Figure 2D–F). In the CAD software, the cement space was set to 120 µm from 1 mm above the preparation finish line (Figure 2D). The designed 3-unit FDPs were saved as standard tessellation language (STL) files. The STL files were loaded into a milling unit (EZIS HM, DDS, Seoul, Republic of Korea) and the 3-unit FDP was fabricated using a lithium disilicate block (IPS e.max CAD, Ivoclar Vivadent, Schaan, Liechtenstein). The milled 3-unit FDPs were crystallized according to the manufacturer's recommended settings by using a ceramic firing furnace (Multimat NTX, Dentsply Sirona) (Figure 3A).

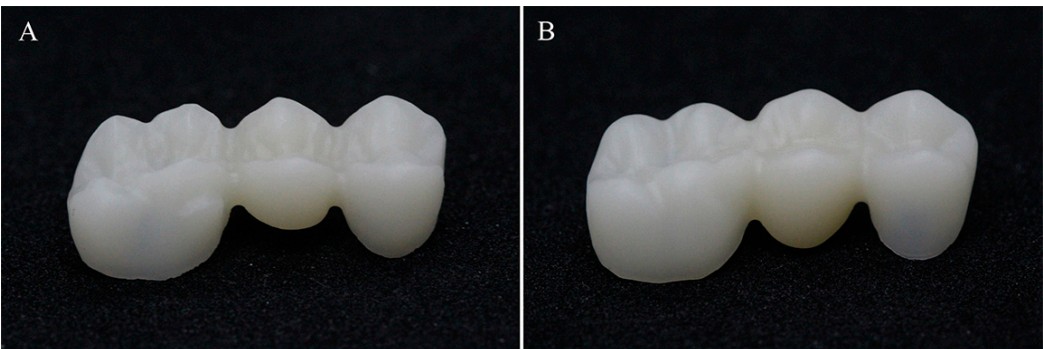

**Figure 3.** Three-unit fixed dental prosthesis fabricated using two digital workflows. (**A**) In-lab group. (**B**) Chairside group.

The chairside (closed type) group used a fully digital workflow. Anti-reflective powder with an approximate thickness of 2 µm was uniformly applied to the metal model and the metal model was scanned using an intraoral scanner (CEREC Omnicam, Dentsply Sirona) according to the manufacturer's recommendations. One investigator (K.S.) performed a total of 15 scans with each intraoral scanner. The obtained virtual models were transferred to the CAD software (CEREC S/W 4.4.4, Dentsply Sirona) in order to design a 3-unit FDP. In the CAD software, the cement space was set to be the same as the in-lab (open type) group. The designed 3-unit FDPs were saved as STL files which were then loaded into a milling unit (CEREC MC XL, Dentsply Sirona) and fabricated using the same material as the in-lab (open type) group. The milled 3-unit FDPs were then crystallized (Figure 3B).

*2.2. Evaluation of Trueness*

To evaluate the trueness of the milling unit, the procedure shown in Figure 4 was carried out. The inner surface of the 3-unit FDP was scanned by using a contact-type scanner (DS10, Renishaw plc, Gloucestershire, UK) (Figure 4A) by one skilled investigator (K.S.) to ensure that the scan did not interfere with the undercut during the contact scan. Prior to use, the contact scanners were verified for accuracy by calibration according to the manufacturer's recommendations. The scanned virtual 3-unit FDP was stored as an STL file (Figure 4B). The designed virtual 3-unit FDP was also stored as an STL file during the CAD process of both the in-lab (open type) and chairside (closed type) groups.

The STL files of the scanned and designed virtual 3-unit FDP were loaded on the 3-dimensional (3D) inspection software (Geomagic Control X, 2018.0.1, 3D Systems, Rock Hill, SC, USA). The designed

virtual 3-unit FDP was set as the reference data and the inner surface was specified based on the FDP margin. The scanned virtual 3-unit FDP was superimposed on the reference data in the order of the initial alignment and best-fit alignment (Figure 4C). The difference values of the superimposed data were analyzed through the 3D compare function in the inspection software. All point clouds within the 3-unit FDP of the reference data and all point clouds within the scanned virtual 3-unit FDP were calculated as the root mean square (RMS) values as follows [13]:

$$RMS = \frac{1}{\sqrt{n}} \cdot \sqrt{\sum_{i=1}^{n} (X_{1,i} - X_{2,i})^2}.$$

For all the data points, $X_{1,i}$ was the position of measurement point No. $i$ in the reference scan data, and $X_{2,i}$ was the position of measurement point No. $i$ in the evaluation scan data. Also, $n$ refers to the total number of data points measured in each analysis. The RMS values show a higher degree of 3D agreement as the value approaches zero.

To show the difference of trueness better, a color difference map was used with a range of ±100 μm (20 color segments) in the 3D inspection software (Figure 3D). In the color map, the red zone (10–100 μm) represents a positive error, meaning that the milling did not reach the target position. The blue zone (−10 to −100 μm) indicates a negative error, meaning that the milling was more than the target position. The green zone (±10 μm) indicates very precise machining accuracy.

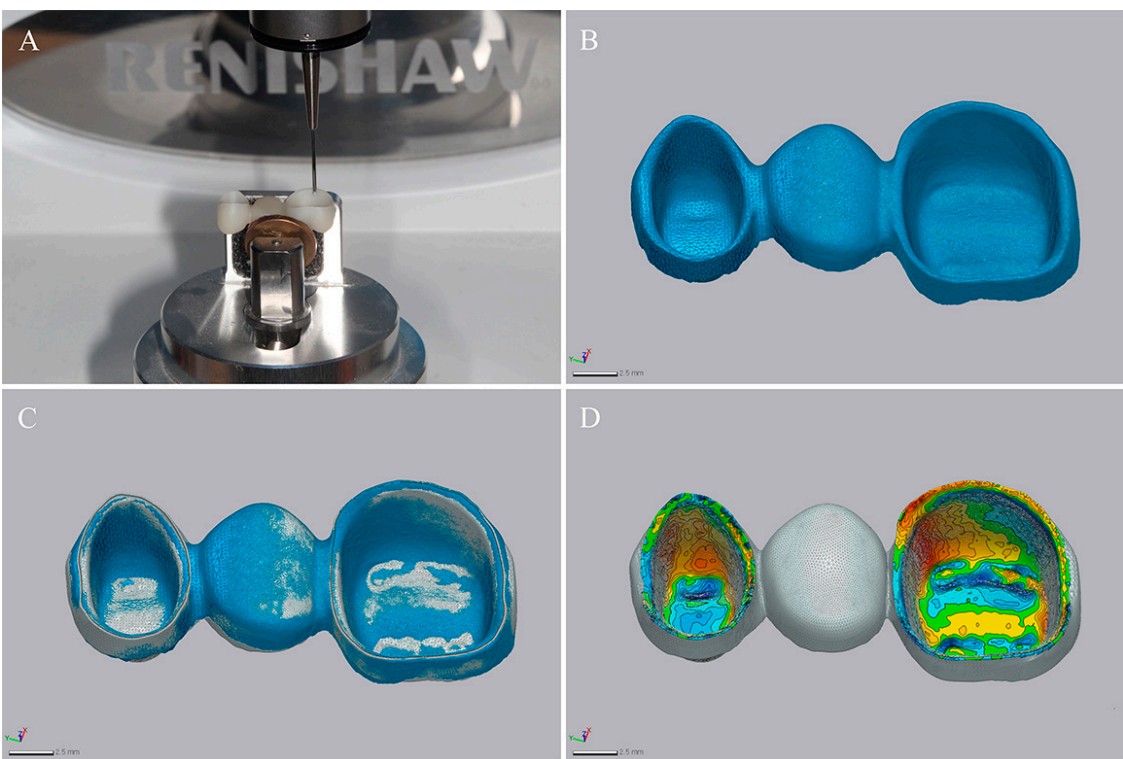

**Figure 4.** Process for the 3-dimensional analysis of trueness. (**A**) Touch-probe scanning. (**B**) Inner surface scan of 3-unit fixed dental prosthesis. (**C**) Superimposition. (**D**) Color difference map.

### 2.3. Evaluation of Fitness

The marginal and internal fit of the 3-unit FDP were evaluated as shown in Figure 5. The inner surface of the 3-unit FDP was filled with light body impression materials and carefully seated on the abutment of the metal model. It was then confirmed that the 3-unit FDP was positioned correctly and lightly loaded with finger pressure. To apply the same load on the 30 specimens, a gauze was placed on the 3-unit FDP and a binder clip was positioned to engage the occlusal surface and the bottom

of the metal model (Figure 5A). The polymerization of the impression material was completed and carefully separated from the abutment so that the impression material did not fall on the inner surface of the 3-unit FDP. Using a desktop optical scanner, the inner surface of 3-unit FDP with (Figure 5B) and without the impression materials (Figure 5C) was scanned by using a desktop optical scanner and stored as STL files. The STL files were loaded on the 3D inspection software (Geomagic Control X) and superimposed in the order of initial alignment and best-fit alignment (Figure 5D). The hypothetical planes passing through the center of the first premolar and the first molar, in the buccolingual and mesiodistal direction, were set (Figure 5E). The hypothetical planes were set in the same position in all the specimens based on the coordinates of the first set plane. The fitness was measured in the margin, chamfer, axial, angle, and occlusal regions (Figure 5E).

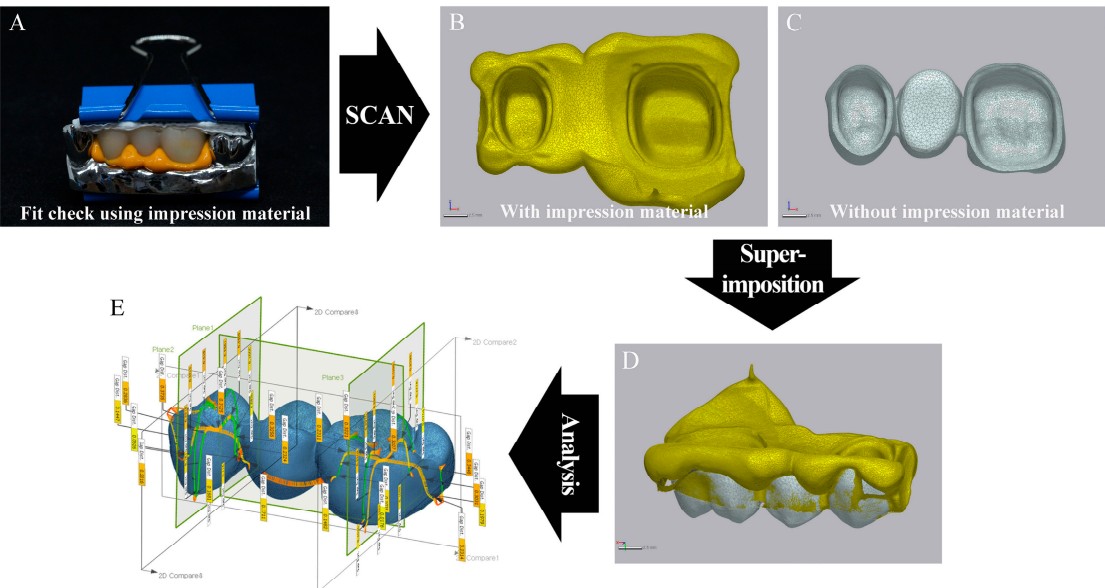

**Figure 5.** Measurements of the marginal and internal fit. (**A**) Fit check using impression materials. (**B**) Inner surface scan of the 3-unit fixed dental prosthesis (FDP) with impression materials. (**C**) Inner surface scan of the 3-unit FDP without impression materials. (**D**) Superimposition. (**E**) Measurements of the marginal and internal fit in a hypothetical plane.

*2.4. Statistical Analysis*

All data were analyzed using SPSS statistical software (release 23.0, IBM, Chicago, IL, USA). First, the normal distribution of data was investigated using the Shapiro–Wilk test. In the non-normal distribution, differences between the groups were analyzed using the Mann–Whitney U-test ($\alpha = 0.05$). The Pearson correlation test was conducted to analyze the relationship between the trueness of the milling unit and the fitness ($\alpha = 0.05$).

## 3. Results

The results showed a significantly lower marginal and internal fit in the in-lab (open type) group at all measurement positions ($p < 0.001$; Figure 6; Table 1). The mean value of marginal fit was significantly different in the in-lab (open type) group (113.3 ± 62.3 μm) and the chairside (closed type) group (210.8 ± 83 μm) ($p < 0.001$).

The trueness of the milling unit was significantly different in the in-lab (open type) group (46.7 ± 11.7 μm) and the chairside (closed type) group (27.7 ± 6.1 μm) ($p < 0.001$; Figure 7; Table 2). Comparing the color difference map of the occlusal region in Figure 8, the in-lab (open type) group shows blue zone (excessive milling) more than red. On the other hand, the chairside (closed type) group shows red zone (insufficient milling) more than blue.

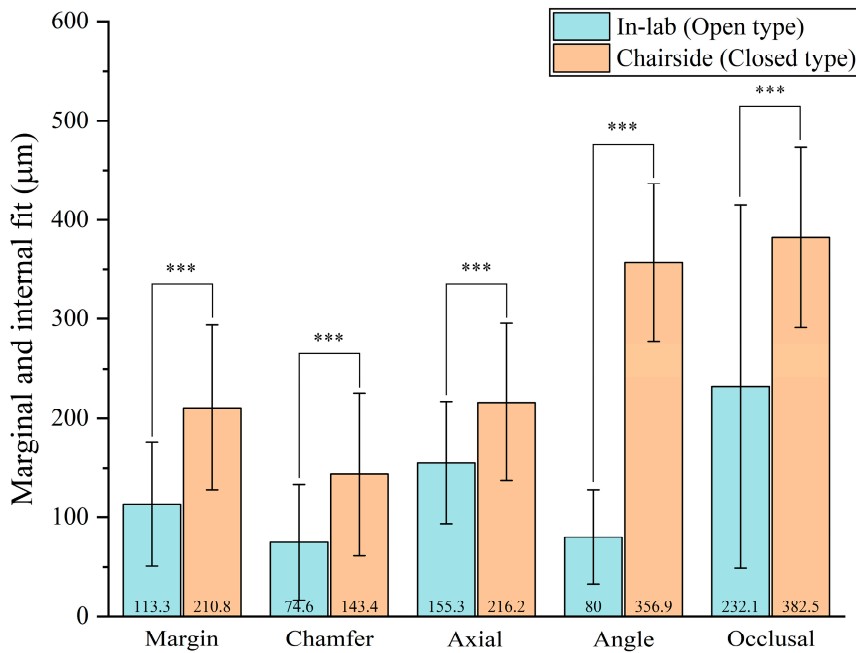

**Figure 6.** Comparison of the marginal and internal fit mean values by digital workflows. ***, Significant (*p* < 0.001).

**Table 1.** Comparison of the marginal and internal fit according to digital workflows.

| System | Margin | Chamfer | Axial | Angle | Occlusal |
|---|---|---|---|---|---|
| | Fitness (µm), Mean ± SD | | | | |
| **Chairside group** | **210.8 ± 83** | **143.4 ± 82.1** | **216.2 ± 79.2** | 356.9 ± 79.6 | 382.5 ± 91.4 |
| In-lab group | 113.3 ± 62.3 | 74.6 ± 58.5 | 155.3 ± 61.9 | 80 ± 47.8 | 232.1 ± 183 |
| *P* | <0.001 *** | <0.001 *** | <0.001 *** | <0.001 *** | <0.001 *** |

RMS: Root mean square. Significance was determined by the *** Mann–Whitney U-test.: *p* < 0.001.

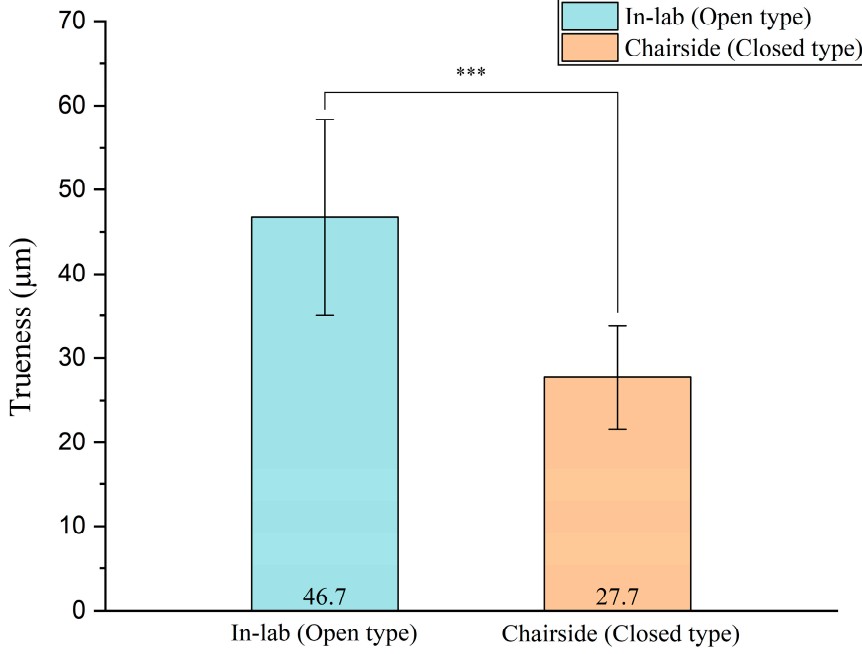

**Figure 7.** Comparison of the trueness mean values by digital workflows. ***, Significant (*p* < 0.001).

**Table 2.** Comparison of the trueness values according to digital workflows.

| System | Trueness (µm), Mean ± SD |
| --- | --- |
| Chairside group | 27.7 ± 6.1 |
| In-lab group | 46.7 ± 11.7 |
| *P* | <0.001 *** |

Significance was determined by the *** Mann–Whitney U-test.: *p* < 0.001

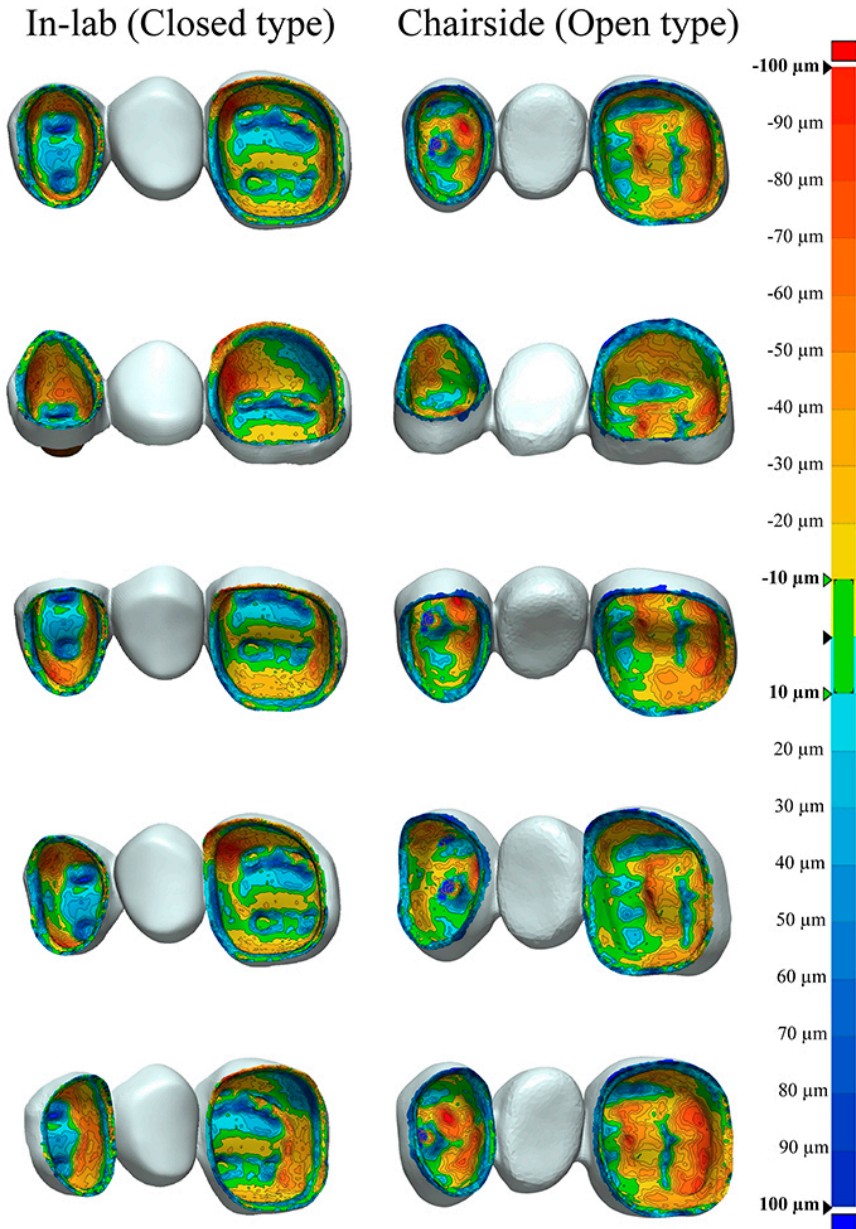

**Figure 8.** Comparison of the color difference map of trueness by digital workflows.

The results of the correlation between the trueness of the milling unit and the fitness are shown in Figure 9. The trueness and internal fit showed a positive correlation (correlation coefficient = 0.621) only in the in-lab (open type) group (*p* = 0.013). There was no significant correlation between the trueness and marginal fit in the in-lab (open type) group (*p* = 0.070). There was no significant correlation

between trueness and marginal fit ($p = 0.855$), as well as the trueness and internal fit ($p = 0.701$), in the chairside (closed type) group.

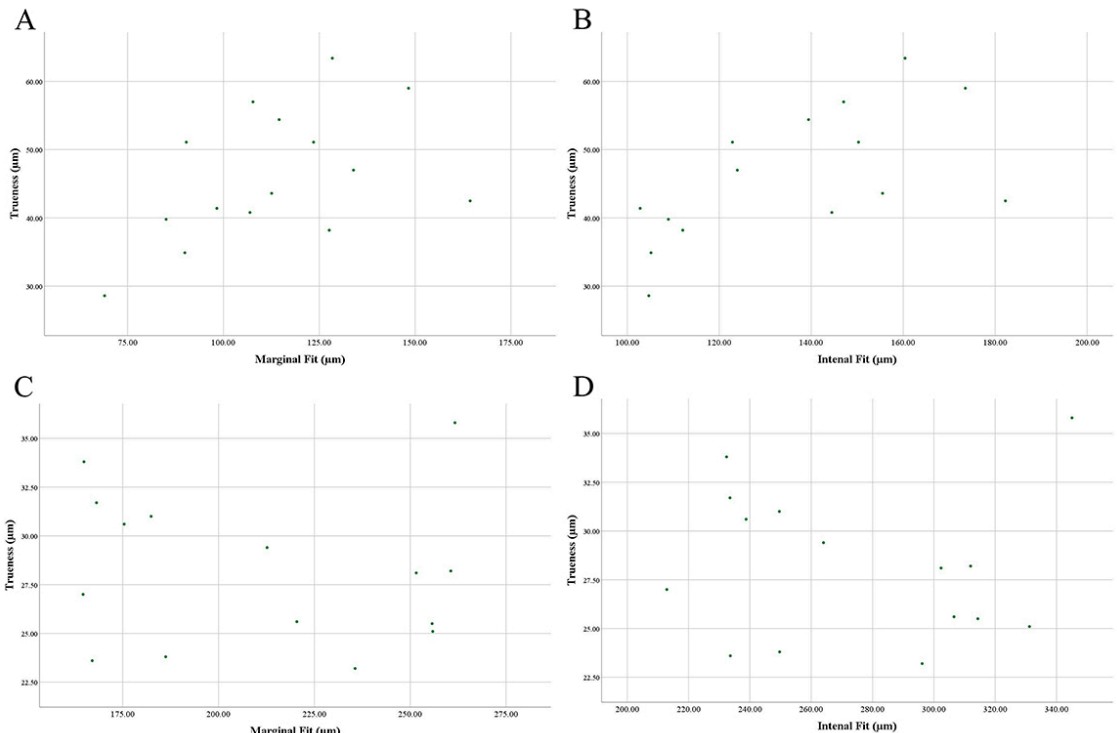

**Figure 9.** Relationships between the trueness of the milling unit and fitness. (**A**) In-lab group (trueness and marginal fit, $p = 0.070$). (**B**) In-lab group (trueness and internal fit, $p = 0.013$). (**C**) Chairside group (trueness and marginal fit, $p = 0.855$). (**D**) Chairside group (trueness and internal fit, $p = 0.701$).

## 4. Discussion

For the purpose of this study, the fitness and trueness of the 3-unit FDP fabricated by two digital workflows and their correlation were evaluated. The first null hypothesis (the 3-unit FDP fabricated by the two digital workflows was not different in the assessment of fitness and trueness) was rejected ($p < 0.001$), while the second null hypothesis (which has no correlation with fitness and trueness) was partially rejected. Based on our findings, it can be seen that the difference in the equipment used in the digital workflow affects the fitness and trueness of the 3-unit FDP. In the same way, previous studies were carried out with different digital workflows according to the equipment used, and there were significant differences according to the workflow [2,11]. In Ortorp's study [26], the digital workflow ($222.5 \pm 124.6$ µm) showed a significantly higher marginal fit than the conventional workflow ($118 \pm 49.7$ µm). The results were similar to the marginal fit ($210.8 \pm 83$ µm) of the chairside (closed type) group presented in this study. Varol [27] also reported a higher marginal fit with the digital workflow ($86.17 \pm 27.61$ µm) than the conventional workflow ($77.26 \pm 29.23$ µm). Bayramoglu [28] also reported a higher marginal fit with the digital workflow ($120.4 \pm 54.5$ µm) than the conventional workflow ($75.4 \pm 16.6$ µm). However, in Berejuk's study [29], the marginal fit was significantly higher in the conventional workflow ($11.56 \pm 8.74$ µm) than in the digital workflow ($1.85 \pm 1.50$ µm). Previous studies have shown different results. According to the systematic review of multiunit FDPs fabricated using digital workflow [25], single FDPs fabricated using the digital workflow have been generally studied, but multiunit FDPs have been insufficiently examined, and further studies are required for clinical reliability.

In this study, the marginal and internal fit of the in-lab (open type) group were significantly lower at all measurement positions ($p < 0.001$). The most significant steps for the fabrication of FDPs are the scan acquisition process and the CAM process [21]. Among them, the scan acquisition process can

have the greatest influence on FDP fitness [30]. Abduo [2] evaluated the accuracy of casts produced using conventional and digital workflows and reported poor accuracy of the digital workflow using intraoral scanners. Park [1] reported that the scan error increases as the range of the intraoral scans increases, and the recommended scanning range from single to 3-unit FDP, according to the intraoral scanner. Thus, the limited scan range of the intraoral scanner is responsible for the poor fit of the 3-unit FDP. In our study, a fully digital workflow was designed using only one manufacturer's product (CEREC). Recently released equipment from different manufacturers can be used in each process [2]. Therefore, additional studies using a combination of recently introduced intraoral scanners, CAD software, and milling units from various manufacturers are needed.

Trueness indicates the similarity between the CAD reference model and the CAD test model [21]. Previous studies have evaluated the trueness of the CAM process [21]. Kim [31] reported excellent trueness when using the three mill burs rather than the two mill burs. Kirsch's study [32] showed excellent trueness using the five-axial milling unit compared to the four-axial milling unit. In this study, two mill burs were used in the four-axial milling unit for both the chairside (closed type) and in-lab (open type) groups, and our findings show that the milling unit influences the trueness. Further studies are required to evaluate the trueness using a various number of mill burs and a five-axial milling unit.

In order to measure the trueness, the reference data must exist and the method of acquisition is very important. In previous studies [33], the trueness was analyzed by calculating the distance from the coordinates of the reference data using a contact type scanner. In other studies [1], the trueness was measured by acquiring the reference data using a precision industrial scanner. Contact scanners were more accurate and stable than optical scanners, and more efficient in reproducing the abutment margins than laser scanners [34]. In previous studies [35], the abutment was scanned with both an optical scanner and a contact scanner, and excellent reproducibility was obtained with a contact scanner than an optical scanner.

The trueness of the outer surface of the crown was not evaluated in this study since the narrow, deep, and undercut regions were not contact-scannable. In addition, when separating the crown from the block, the external connections may affect trueness. However, further research is needed to analyze the trueness of the outer surface of the crown.

In this study, the correlation between the trueness of the milling unit and the fitness showed that trueness and internal fit were correlated only in the in-lab (open type) group ($p = 0.013$) (Figure 9), while there was no correlation between the trueness and internal fit in the chairside (closed type) group ($p > 0.05$). The reason for this is that the trueness of the in-lab (open type) group was significantly poorer than the chairside (closed type) group ($p < 0.001$). Therefore, it can be assumed that the poor trueness of the milling unit adversely affects the internal fit.

## 5. Conclusions

Based on the findings of this in vitro study, significant differences were observed in the marginal and internal fit according to the digital workflows. In-lab (open type) digital workflows showed an excellent marginal and internal fit compared to the chairside (closed type) group. The use of proper equipment in the in-lab (open type) digital workflow makes it possible to make better 3-unit FDPs than the chairside (closed type) digital workflow. The poor trueness values of the milling unit have an adverse effect on the internal fit. For this reason, the most significant steps for the fabrication of FDPs are the scan acquisition process and the CAM process.

**Author Contributions:** Conceptualization, K.S.; methodology, D.J.; validation, K.-b.L.; formal Analysis, K.S.; investigation, K.S.; data curation, D.J.; writing—original draft, D.J. and K.S.; visualization, K.S.; supervision, K.-b.L.; project administration, K.-b.L.; Revised the paper, K.S.

**Funding:** This research was supported by the Ministry of Trade, Industry & Energy (MOTIE, Korea) under the Industrial Technology Innovation Program (No. 10062635); and the Institute for Information & Communications Technology Promotion (IITP) for a grant funded by the Korea government (MSIP) (B0101-19-1081); and Korea Institute for Advancement of Technology (KIAT) through the National Innovation Cluster R&D program (P0006691).

**Acknowledgments:** The authors thank the researchers of the Advanced Dental Device Development Institute, Kyungpook National University for their time and contributions to the study.

**Conflicts of Interest:** The authors declare no conflicts of interest. The funders had no role in the design of the study; in the collection, analyses, or interpretation of the data; in the writing of the manuscript, or in the decision to publish the results.

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
