# Peer review of "A Comparative Study of the Fitness and Trueness of a Three-Unit Fixed Dental Prosthesis Fabricated Using Two Digital Workflows"

_applsci, doi:10.3390/app9142778_

Round 1
Reviewer 1 Report
After reading the paper entitled:A comparative study of the fitness and trueness of a 3-unit fixed dental prosthesis fabricated using two digital workflows,
I have to agree with the authors that the difference in the equipment used in the digital workflow affects the fitness and trueness of the 3-unit FDP. As I noticed from the references, the authors have published related articles in the past and, probably, some will follow.
In my opinion it would have been of interest to insert a few images with the Co-Cr printed model and the scanning/digital design phases.
A few errors in the text need correction:
Line 201. correlation coefficient=.621. Please correct this.
Line 223 Bayramoglu28. Please correct this.
Author Response
We are grateful to the reviewers for their critical comments and useful suggestions that have helped us to greatly improve our paper. As indicated in the following responses, we have reflected all these comments in the revised version of our paper.
Reviewer #1
1. In my opinion it would have been of interest to insert a few images with the Co-Cr printed model and the scanning/digital design phases.
Response: We very much appreciate the reviewer’s comment and respect the reviewer’s insight. We have carefully considered your comments. As the reviewer notes, we have added figure 2 about Scan process of model, and design process of three-unit fixed dental prosthesis.
2. A few errors in the text need correction:
Line 201. correlation coefficient=.621. Please correct this.
Line 223 Bayramoglu28. Please correct this.
Response: Thank you for your suggestion for improving the quality of the manuscript. We have revised the issue that you point out.

Reviewer 2 Report
The study compares two digital workflows on the fitness and trueness of a 3-unit dental prothesis.
Abstract: Kindly write “2” as “two” and adjust throughout the whole mansucript, as well as “5” in line 83.
The introduction is well written and involves all necessary information.
Materials and methods:
- This part is also well described with all necessary details mentioned! I would suggest to add subheadings as follows:
Line 83: 2.1. sample preparation
Line 125: 2.2. evaluation of trueness
Line 154: 2.3. evaluation of fitness
Line 174: 2.4. statistical analysis
- line 140: kindly add a reference to your equation.
Conclusions: Kindly add the statement from line 231-232 that the most significant steps for the fabrication of FDPs are the scan acquisition process and the CAM process, as this statement is of greatest importance to the clinician. Thank you for the good manuscript!
Author Response
We are grateful to the reviewers for their critical comments and useful suggestions that have helped us to greatly improve our paper. As indicated in the following responses, we have reflected all these comments in the revised version of our paper.
Reviewer #2
1. Abstract: Kindly write “2” as “two” and adjust throughout the whole mansucript, as well as “5” in line 83.
Response: Thank you for your suggestion for improving the quality of the manuscript. We have revised the issue that you point out.
2. This part is also well described with all necessary details mentioned! I would suggest to add subheadings as follows:
Line 83: 2.1. sample preparation
Line 125: 2.2. evaluation of trueness
Line 154: 2.3. evaluation of fitness
Line 174: 2.4. statistical analysis
Response: We very much appreciate the reviewer’s comment and respect the reviewer’s insight. We have added the subheadings.
3. line 140: kindly add a reference to your equation.
Response: Thank you for your suggestion for improving the quality of the manuscript. We have added the reference.
4. Conclusions: Kindly add the statement from line 231-232 that the most significant steps for the fabrication of FDPs are the scan acquisition process and the CAM process, as this statement is of greatest importance to the clinician.
Response: We very much appreciate the reviewer’s comment and respect the reviewer’s insight. We have carefully considered your comments. We have added the statement that you point out.
